# Radiography and Computed Tomography Detection of Intimal and Medial Calcifications in Leg Arteries in Comparison to Histology

**DOI:** 10.3390/jpm12050711

**Published:** 2022-04-29

**Authors:** Annelotte Vos, Aryan Vink, Remko Kockelkoren, Richard A. P. Takx, Csilla Celeng, Willem P. T. M. Mali, Ivana Isgum, Ronald L. A. W. Bleys, Pim A. de Jong

**Affiliations:** 1Department of Pathology, University Medical Center Utrecht and Utrecht University, 3584 CX Utrecht, The Netherlands; a.vos-15@umcutrecht.nl (A.V.); a.vink@umcutrecht.nl (A.V.); 2Department of Pathlogy, Meander Medical Center, 3800 BM Amersfoort, The Netherlands; 3Department of Radiology, University Medical Center Utrecht and Utrecht University, 3584 CX Utrecht, The Netherlands; remkokockelkoren@gmail.com (R.K.); richard.takx@gmail.com (R.A.P.T.); celengcsilla@gmail.com (C.C.); w.mali@umcutrecht.nl (W.P.T.M.M.); 4Department of Radiology and Nuclear Medicine, Amsterdam UMC, 1105 AZ Amsterdam, The Netherlands; ivana.isgum@gmail.com; 5Department of Anatomy, University Medical Center Utrecht and Utrecht University, 3584 CX Utrecht, The Netherlands; r.l.a.w.bleys@umcutrecht.nl

**Keywords:** vascular calcification, computed tomography, radiography, histology, medial arterial calcification, atherosclerosis

## Abstract

Calcifications are common in the tunica intima and tunica media of leg arteries. There is growing interest in medial arterial calcifications, as they may be modifiable with treatment. We aimed to investigate radiography and computed tomography (CT) for the detection and characterization of both types of arterial calcification in leg arteries in relation to histology. In a postmortem study we therefore investigated 24 popliteal and 24 tibial arteries. The reference standard was presence of arterial calcification and the dominance of intimal or medial calcification on histology. Radiographs and CT scans were scored for presence of calcification and for dominant intimal or medial pattern based on prespecified criteria (annularity, thickness, continuity). Both radiography and CT detected 87% of histologically proven calcifications but missed mild calcifications in 13%. When only the arteries with detected calcifications were included, a moderate agreement was observed on intimal/medial location of calcifications between histology and radiography (correct in 19/24 arteries (79%); Kappa 0.58) or CT (correct in 33/46 arterial segments (72%); Kappa 0.48). With both modalities there was a slight tendency to classify intimal calcifications as being located in the media and to miss media calcification. Our study demonstrates the potential and limitations of both radiography and CT to detect and classify arterial calcifications in leg arteries.

## 1. Introduction

Critical limb threatening ischemia of the lower extremities is a common problem with a prevalence depending on age and diagnostic criteria [1]. Lower extremity vascular calcification is common [2] and the calcifications can be located in tunica intima and the tunica media [3,4,5]. The role of these calcifications in the development of ischemia is uncertain. It has been suggested that medial arterial calcification is a cause of arterial stiffening and events [6,7]. If the calcification process indeed contributes to ischemia, these calcifications could be an interesting therapeutic target. Several companies are now aiming to develop drugs for patients with genetic syndromes, chronic renal disease, and beyond. In particular, medial arterial calcification is thought to be potentially modifiable and these calcifications are more common in patients with rare genetic syndromes, diabetes, chronic renal failure, and aging [8,9,10].

Further scientific investigation of the role of calcification in peripheral arterial disease and critical limb threatening ischemia detection and distinction of these calcifications is important, also in vivo. The literature on radiological techniques for this purpose is limited, but in the past, radiographic criteria were developed to distinguish intimal and medial calcification in leg arteries on radiographs [11]. Recently, for carotid arteries, similar criteria were able to distinguish both types of calcification on computed tomography (CT) to some extent [12,13]. The aim of this original research study was to investigate the ability of radiography and CT to detect and classify calcification in the peripheral arteries of the lower extremity in a postmortem study.

## 2. Materials and Methods

The popliteal and posterior tibial artery were both examined at two predetermined locations in 24 lower extremities by radiography, CT and histology. After the radiographs and scans were acquired, we harvested the popliteal artery tissue at 8–9 and 13–14 cm proximal of the origin of the anterior tibial artery. For the posterior tibial artery, we harvested tissue 8–9 and 13–14 cm distal of the origin of the anterior tibial artery. The leg specimens were derived from 14 individuals aged 70–96 years who donated their bodies for research and education. Written informed consent was obtained during their lives. Data on medical history were not available. 

Plain radiographs (anterior–posterior and lateral) and non-enhanced CT-scans were obtained from all specimens. The CT scans were acquired on the iCT scanner (Philips Healthcare, Cleveland, OH, USA). The voltage was set at 80 kVp and the amperage at 50 mAs. Slice thickness was 0.9 mm. The radiographs were acquired by using the DRX revolution (Carestream) at 70 kV and 8 mAs in both anteroposterior and lateral directions. On the radiographs, calcifications in the arteries were divided into absent, intimal (patchy, thick, irregular calcifications), or medial (continuous, regular, thin, tubelike calcifications), based on the criteria from Orr et al [11]. Given the difficulty in measuring the distance from the tibial artery origin on radiographs, the complete popliteal and tibial arteries were scored on the radiographs and not the separate arterial segments. These results were compared to the average of the histology results. For CT, points were given for cross-sectional thickness (absent [0 points], >1.5 mm thick [1 point], or <1.5 mm thin [3 points]) and annularity (absent [0 points], dot(s) [1 point], <90 degrees [2 points], 90–270 degrees [3 points], or >270 degrees [4 points]) of calcifications. On the longitudinal axis, morphology of the calcification was scored (absent [0 points], indistinguishable [0 points], patchy [1 point], or continuous [4 points]) [12]. As previously proposed, intimal dominance was defined as <7 points and medial dominance as ≥7 points [12]. All radiographs and CT scans were scored by two independent observers. One was a board-certified radiologist with over 10 years of experience in reading radiography and CT and the other a first-year resident with a PhD in coronary imaging. Both were blinded for the histology results. Radiographs were scored blinded for the CT results and vice versa. The scores of both observers were used to determine inter-rater reliability, the scores of the most experienced observer were compared with the histological data. 

After pressure formaldehyde (4%) fixation of the bodies, a total of 24 popliteal and 24 tibial arteries were collected. The arteries were decalcified using diaminoethylene tetra-acetic acid for at least 48 h. Decalcification was necessary for maintenance of the vascular architecture and did not influence the analysis since evaluation of calcification is based on visualization of matrix previously altered by the calcification process [14,15]. Two segments were obtained from each artery; 1 cm segments of both the tibial artery and popliteal artery were collected 8 and 13 cm distal and proximal of the origin of the anterior tibial artery, respectively. The harvested segments were divided into samples of 3–4 mm, perpendicular to the lumen. On digitalized (Scan-Scope XT scanner, Aperior Technologies) microscopic hematoxylin and eosin and elastic van Gieson stained slides, the surface of vascular calcification in the intima and media was measured using Aperio Image Scope software (Aperio Technologies, Vista, CA, USA). Calcifications around the internal elastic lamina were classified as medial arterial calcifications [16]. An average per segment of 1 cm was calculated to compare with the radiology results. Calcifications were categorized in intimal or medial dominance based on the largest surface of calcification. Only calcifications of >1% of the vascular surface were scored as positive for calcification. 

Inter-rater and inter-method reliability were determined using Cohen’s Kappa with 95% confidence intervals. In case of ordinal measures (thickness, annularity) linear weighted Kappa coefficients were used. A Kappa value of 0.41–0.60 was regarded as moderate and a value >0.61 as substantial agreement [17]. Data were analyzed using SPSS version 25.0 (IBM Corporation, New York, NY, USA). 

## 3. Results

### 3.1. Histology

A total of 96 1 cm long arterial segments (48 tibial and 48 popliteal) were histologically investigated. 

To match histology with radiography, the histology results of the 96 samples were averaged (mean of two segments per artery) resulting in 48 arteries (24 tibial and 24 popliteal arteries) available for analysis. No calcification was histologically present in 18 (38%) arteries; in 12 (25%) arteries, 1–5% of the arterial wall was calcified on average; in 11 (23%) arteries, 5–25% was calcified on average; and in 7 (15%) arteries, ≥25% was calcified.

To match histology with CT, all 96 arterial segments were investigated separately. In 38 of 96 (40%) segments, no calcifications were histologically present. In 22 (23%) segments, 1–5% of the vascular wall was calcified; in 24 (25%), 5–25% was calcified; in 11 (11%), 25–50% was calcified; and in one segment (1%), ≥50% was calcified.

### 3.2. Radiography

On the radiographs, 48 arteries (24 tibial and 24 popliteal arteries) were categorized as intimal, medial, or no calcification based on the described criteria (Figure 1) [11]. In 18/48 arteries both histology and radiology showed no calcifications. In 24/48 arteries, both histology and radiology showed the presence of some type of calcification. In 6/48 (13%) arteries, calcifications which were not seen on radiographs were observed histologically (Table 1). In all six arteries where calcifications were not seen on radiograph, only small amounts of calcification (1–6% of the vascular wall surface calcified) were present histologically. 

Overall, when comparing the experienced radiologists with the histologic dominant pattern 37/48 (77%) arteries were correctly classified as no calcification, intimal, or medial calcification (Table 1). When only the arteries with both histologically and radiologically detected calcifications were included, 19/24 (79%) arteries were correctly classified as predominantly intimal or predominantly medial calcification (Cohen’s Kappa 0.58 (95% CI 0.29-0.88); Table 1). There was a tendency to classify intimal as medial calcification with this method; in 5/24 (21%) arteries with calcification, intimal calcification was incorrectly classified as medial (Table 1). 

The inter-rater reliability between the experienced and inexperienced observer was poor (Cohen’s Kappa 0.17 (95% CI 0.01–0.33); Table 2).

### 3.3. Computed Tomography

In 96 arterial segments (48 of the tibial artery and 48 of the popliteal artery), the different characteristics of calcifications (thickness, annularity and continuity) were scored and classified as no calcification, predominantly intimal or predominantly medial calcification, or indistinguishable (Figure 1) [12]. In 34/96 arterial segments, no calcifications were seen both histologically and on CT. In 12/96 (13%) arterial segments, the histologically observed calcifications were not detected on CT. In most of these cases, only small amounts of calcification were present (1–6% of the vascular wall surface calcified), however, in three cases, large amounts of calcification (23–28% of the vascular wall surface calcified) was missed, and in four cases, calcifications were seen on CT while histology showed no calcifications, representing a potential problem with exact matching between histology and CT.

Overall inter-method reliability between the experienced observer and histology was moderate (Cohen’s Kappa 0.48 (95% CI 0.26–0.70)), with 33/46 (72%) of the arterial segments with calcifications seen in both modalities correctly classified (Table 1). In 6/46 (13%) cases, intimal calcification was radiologically classified as medial calcification and in 3/46 (7%), medial calcification was radiologically classified as intimal calcification. In 4/46 (9%) cases, calcification was radiologically unclassifiable. 

The interrater reliability between the experienced and inexperienced observer for overall classification of the calcifications was better than for radiography, but still poor (Cohen’s Kappa 0.39 (95% CI 0.27–0.52); Table 3). When comparing the different characteristics, good interrater reliability for annularity (linear weighted Kappa 0.69 (95% CI 0.59–0.80)) and thickness (linear weighted Kappa 0.69 (95% CI 0.58–0.81)) was seen, but morphology was more difficult to score (Cohen’s Kappa 0.36 (95% CI 0.34–0.61)). 

## 4. Discussion

Critical limb-threatening ischemia is a devastating disease, and the role of arterial wall calcification remains poorly understood. Arterial calcifications can occur in both the atherosclerotic intimal and medial layer of the vascular wall in the lower extremities, and both may influence the risk of ischemia [6]. There is growing interest in the treatment of medial arterial calcifications and a variety of compounds was recently reviewed [18]. For investigations into the role of calcification, the in vivo detection and characterization of both types of calcification is important. The aim of this original research study was to investigate the ability of radiography and CT to detect and classify calcification in the peripheral arteries of the lower extremity in a postmortem study.

With radiography 77% and with CT 70% of the samples were correctly classified as no calcification, intimal, or medial calcification, and most errors (13%) were detection errors due to small amounts of calcifications or due to possible matching errors. A correct classification of the predominant localization of calcifications in arteries/arterial segments with calcification was found in 79% (radiography) and 72% (CT) of samples. Our results show the possibility of both radiography and CT to detect and distinguish intimal and medial arterial calcification based on morphological criteria in large epidemiological studies. 

A previous study in multiple arteries (aorta, iliac, femoral, popliteal, tibial, superior mesenteric, celiac, splenic) found a correct classification of the localization of calcifications of 36/39 arterial samples (92%) [11]. However, in this study, only heavily calcified samples were selected. The previous study describing the development of the CT-scoring model for distinguishing intimal and medial arterial calcification found a correct classification in 50–56% of the arteries, while no classification could be given in 19–25%, and 25% of the arteries were incorrectly classified [12]. The percentage of incorrect classified samples is comparable to the results of our study, while in our study, more often a distinction between intimal and medial arterial calcification could be made radiologically.

Our study shows poor inter-rater reliability except for annularity and thickness on CT, while the previous study describing the development of the CT-scoring model in the intracranial internal carotid artery found good inter-rater reliability with Kappa values of 0.71–0.73 [12]. However, for the experienced observer we found moderate inter-method reliability in comparison with histology. This suggests that in the categorization of calcifications based on morphological criteria a (substantial) learning curve may be present. Another study with more observers and in vivo data would be needed to test this hypothesis. In the present study, rating was more difficult due to the postmortem status, with tissue alterations and sometimes intravascular air. Results may be better on radiographic and CT images acquired during life.

An important limitation of this study is the use of postmortem specimens. Both fixation and the postmortem interval might result in imaging artifacts (e.g., intravascular air) which hamper the assessment of calcifications. Furthermore, due to differences in resolution and slice thickness, it was difficult to exactly match CT and histology slides. Moreover, on radiography, matching was impossible, and the average of the histology results per artery was used to compare with the radiology results. This probably hampers the diagnosis of intimal calcification most due to the patchy nature of the disease. The study was also limited by the use of donor tissue that was anonymized, except age. It would have been of interest to know the medical history (diabetes mellitus, renal failure) of our donors, as these diseases are associated with medial arterial calcification. Unfortunately, this information was lacking. Nevertheless, we still believe the donor tissue was useful for validation of radiography and CT.

Based on the results of our study, we suggest caution with the use of radiologic classification models for intimal and medial arterial calcification in individual patients and in routine care. Future studies may investigate the combination of both morphological criteria and other tests—such as the pattern [18], F-sodium fluoride uptake [19], or ankle-brachial index [20]—and CT/MR angiography to optimize the classification. Additionally, ultrasound may be of use. Scoring methods to assess medial arterial calcification using ultrasound, which does not use ionizing radiation, are being developed [21]. Although the legs are fairly radio-insensitive, there is some risk associated. In our experience, an in vivo unenhanced CT scan of both legs with acquisition parameters of 120 kVp and 150 mAs (*CTDI_vol_ of 10 mGy*) provides good image quality at a dose of around 1 mSv for a 70 kg adult. Based on our study, we believe it would be justifiable to use the current radiography or CT classification by experienced observers in large epidemiological studies.

## Figures and Tables

**Figure 1 jpm-12-00711-f001:**
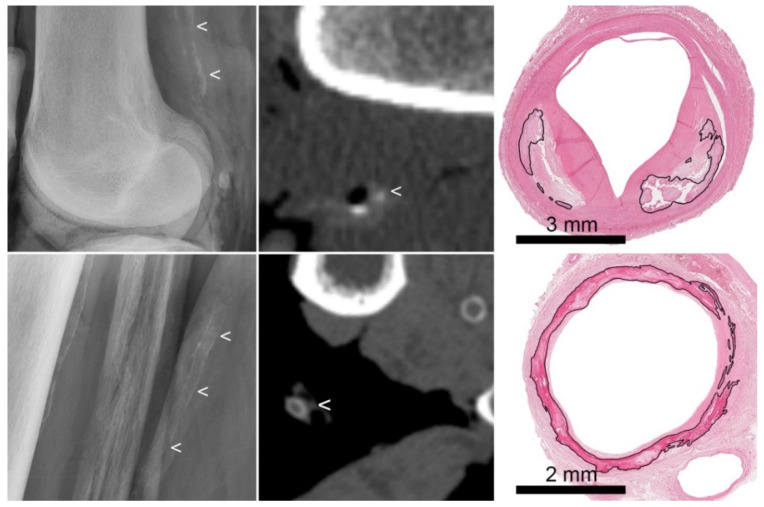
Examples of intimal and medial calcification on radiograph and CT. (**Top row**): atherosclerotic intimal calcifications. From left to right: radiograph (showing irregularly distributed thick calcifications (<)), CT (showing thick dots of calcification (<)), and histology (showing calcifications (marked) located in an atherosclerotic plaque). (**Bottom row**): medial calcifications. From left to right: radiograph (showing regularly distributed thin calcifications along the vascular wall (<)), CT (showing circular thin calcifications (<)), and histology (showing circular calcifications (marked) involving the internal elastic lamina, in the absence of atherosclerosis).

**Table 1 jpm-12-00711-t001:** Inter-method agreement for both radiograph and CT for the experienced observer.

		Histology		
		Absent	Intima	Media
**Radiography**	**Absent**	18	1	5
	**Intima**	0	12	0
	**Media**	0	5	7
**CT**	**Absent**	34	3	9
	**Intima**	3	18	3
	**Media**	0	6	15
	**Indistinguishable**	1	2	2

**Table 2 jpm-12-00711-t002:** Inter-observer agreement for radiography.

		Resident		
		Absent	Intima	Media
**Experienced observer**	**Absent**	9	2	13
	**Intima**	0	2	10
	**Media**	0	3	9

**Table 3 jpm-12-00711-t003:** Inter-observer agreement for CT.

		Resident			
		Absent	Intima	Media	Indistinguishable
**Experienced observer**	**Absent**	31	10	3	2
	**Intima**	0	9	10	5
	**Media**	0	5	15	1
	**Indistinguishable**	0	5	0	0

## Data Availability

The available data are presented in the tables in the manuscript.

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
