# Peer review of "Radiography and Computed Tomography Detection of Intimal and Medial Calcifications in Leg Arteries in Comparison to Histology"

_jpm, 2022, doi:10.3390/jpm12050711_

Round 1

Reviewer 1 Report

The study is interesting, complete and properly done.

There is actually very little to change or modify. 

I appreciate the completeness and the extensiveness of the analysis.

Topic: interesting and timely, of major clinical relevance.

Design: proper and robust for the purpose of the study.

Introduction: complete and exhaustive; very clear and on point.

Methods: complete and exhaustive; very clear and on point.

Results: complete and exhaustive; very clear and on point.

Discussion: complete and exhaustive; very clear and on point.

Limitations: complete and exhaustive; very clear and on point.

Conclusions: complete and exhaustive; very clear and on point.

References: complete and exhaustive; very clear and on point.

Tables: complete and exhaustive; very clear and on point.

Figures: complete and exhaustive; very clear and on point.

Author Response

We thank the reviewer for the time and effort. We are thankful for the positive evaluation. We did a spelling and grammar check and uploaded our revised manuscript with track changes in red letters.

Reviewer 2 Report

The authors must emphasize the original contribution and the clinica aplication of their results

Author Response

We thank the reviewer for the time and effort. We are thankful for the positive evaluation. We did a spelling and grammar check and uploaded our revised manuscript with track changes in red letters. We improved our introduction and discussion based on your feedback and added more text on the clinical relevance and the value of our original contribution.

Reviewer 3 Report

This is an interesting study showing discrepancies between radiologic assessment vs. histologic assessment of intimal and medial arterial calcification in peripheral arteries.

The manuscript is innovative, and well-written and all Ethical disclosures are well presented.

I have no major objections, only would advise authors to proofread this paper and make changes to English syntax accordingly

Author Response

(The authors gave the same response as above.)

Reviewer 4 Report

The manuscript titled "Radiography and computed tomography detection of intimal and medial calcifications in leg 2 arteries in comparison to histology" presents a result of the experimental study. There are some issues that must be addressed:

  1. There is no information about acquisition parameters both for radiography and CT – that has to be added.
  2. No information about the used x-ray machine and CT scanner.
  3. There is a lack of demographic data about donors.
  4. Some disses affect the distribution of calcifications. Is there any donner suffering from diabetes, end-stage kidneys failure, or any other disses known to cause excessive calcification in arteries?
  5. provide some examples of therapy available nowadays that can be applied if the calcification is detected - to prove that investigating that subject can have any impact on the treatment of the patient or it is just to know that he/she has calcification
  6. what was the radiation dose from both examinations which is significant factor for living patients, especially since Doppler US is the gold standard for assessment of peripheral arteries? 

Author Response

Response: We thank the reviewer for the time and effort. We are thankful for the thorough evaluation. We did a spelling and grammar check and uploaded our revised manuscript with track changes in red letters. We improved our introduction, methods and discussion based on your feedback. Our response to your concerns is provided below.

Question 1: There is no information about acquisition parameters both for radiography and CT – that has to be added.

Response: We agree that this is important methodological information. We added the acquisition parameters for radiography and CT and a paragraph to the discussion.

“The CT scans were acquired on the iCT scanner (Philips Healthcare, Cleveland, Ohio). The voltage was set at 80 kVp and the amperage at 50 mAs. Slice thickness was 0.9mm. The radiographs were acquired by using the DRX revolution (Carestream) at 70 kV and 8 mAs in both antero-posterior and lateral direction.”

Question 2: No information about the used x-ray machine and CT scanner.

Response: We agree that this is important methodological information. For details we refer to our previous response.

Question 3: There is a lack of demographic data about donors.

Response: We agree that this information is lacking. The donor tissue however was anonymized. In general the donors are of old age, but we do not have the medical information. We added this as a limitation to our revised manuscript and added more data on the relevance of such medical knowledge.

The study was also limited by the use of donor tissue that was anonymized, except age. It would have been of interest to know the medical history (diabetes mellitus, renal failure) of our donors as these diseases are associated with medial arterial calcification. Unfortunately this information was lacking. Nevertheless we still think the donor tissue was useful for validation of radiography and CT.

Question 4: Some disses affect the distribution of calcifications. Is there any donner suffering from diabetes, end-stage kidneys failure, or any other disses known to cause excessive calcification in arteries?

Response: We agree with the reviewer that this information is interesting, but unfortunately lacking for our donors. See our previous response.

Question 5: provide some examples of therapy available nowadays that can be applied if the calcification is detected - to prove that investigating that subject can have any impact on the treatment of the patient or it is just to know that he/she has calcification

Response: We added this to the discussion including a recent excellent review of therapeutic compounds.

There is growing interest in the treatment of medial arterial calcifications and a variety of compounds was recently reviewed.18

Question 6: what was the radiation dose from both examinations which is significant factor for living patients, especially since Doppler US is the gold standard for assessment of peripheral arteries? 

Response: As we imaged ex vivo and one leg the dose of our validation study cannot be translated directly to in vivo scanning where two legs are simultaneously in the CT gantry. We added to the discussion a proposal for a CT scanning protocol including an estimation of the effective dose. We think there is some debate on the status of ultrasound for measuring intimal and medial arterial calcification although it is indeed the most widely used method and without radiation.

Also ultrasound maybe of use. Scoring methods to assess medial arterial calcification using ultrasound are being developed22 and it does not use ionizing radiation. Although the legs are fairly radio insensitive there is some risk associated. In our experience an in vivo unenhanced CT scan of both legs with acquisition parameters of 120 kVp and 150 mAs (CTDIvol of 10 mGy) provides good image quality at a dose of around 1 mSv for a 70 kg adult.